# Oxidative Inactivation of the Proteasome Augments Alveolar Macrophage Secretion of Vesicular SOCS3

**DOI:** 10.3390/cells9071589

**Published:** 2020-06-30

**Authors:** Mikel D. Haggadone, Peter Mancuso, Marc Peters-Golden

**Affiliations:** 1Graduate Program in Immunology, University of Michigan Medical School, Ann Arbor, MI 48109, USA; haggamd@umich.edu (M.D.H.); pmancuso@umich.edu (P.M.); 2Division of Pulmonary and Critical Care Medicine, University of Michigan Medical School, Ann Arbor, MI, 48109, USA; 3Department of Nutritional Sciences, University of Michigan School of Public Health, Ann Arbor, MI 48109, USA

**Keywords:** alveolar macrophage (AM), extracellular vesicles (EVs), suppressor of cytokine signaling 3 (SOCS3), reactive oxygen species (ROS), proteasome

## Abstract

Extracellular vesicles (EVs) contain a diverse array of molecular cargoes that alter cellular phenotype and function following internalization by recipient cells. In the lung, alveolar macrophages (AMs) secrete EVs containing suppressor of cytokine signaling 3 (SOCS3), a cytosolic protein that promotes homeostasis via vesicular transfer to neighboring alveolar epithelial cells. Although changes in the secretion of EV molecules—including but not limited to SOCS3—have been described in response to microenvironmental stimuli, the cellular and molecular machinery that control alterations in vesicular cargo packaging remain poorly understood. Furthermore, the use of quantitative methods to assess the sorting of cytosolic cargo molecules into EVs is lacking. Here, we utilized cigarette smoke extract (CSE) exposure of AMs as an in vitro model of oxidative stress to address these gaps in knowledge. We demonstrate that the accumulation of reactive oxygen species (ROS) in AMs was sufficient to augment vesicular SOCS3 release in this model. Using nanoparticle tracking analysis (NTA) in tandem with a new carboxyfluorescein succinimidyl ester (CFSE)-based intracellular protein packaging assay, we show that the stimulatory effects of CSE were at least in part attributable to elevated amounts of SOCS3 packaged per EV secreted by AMs. Furthermore, the use of a 20S proteasome activity assay alongside treatment of AMs with conventional proteasome inhibitors strongly suggest that ROS stimulated SOCS3 release via inactivation of the proteasome. These data demonstrate that tuning of AM proteasome function by microenvironmental oxidants is a critical determinant of the packaging and secretion of cytosolic SOCS3 protein within EVs.

## 1. Introduction

Tissue homeostasis is maintained by the dynamic regulation of information transfer amongst cells. In addition to the contributions of direct cell-cell contact and the release of soluble mediators, the importance of extracellular vesicles (EVs) as vectors for the local and systemic transfer of diverse molecular cargoes between cells has become increasingly appreciated [1,2]. Classically, EVs have been classified into exosomes (Exos) and microvesicles (MVs) based on differences in mode of biogenesis, size, and the presence of specific molecular cargoes. However, substantial overlap in size and molecular characteristics is now recognized to limit the ability to categorically differentiate these two classes of EVs when using traditional isolation methods [3,4].

Under normal conditions, the pulmonary alveolar space is characterized by uniquely high oxygen tension. Furthermore, reactive oxygen species (ROS)—whose generation is amplified during inflammatory responses to inhaled pathogens, antigens, and xenobiotics—are well known to promote pro-inflammatory signaling pathways including the interleukin (IL)-6/signal transducer and activator of transcription 3 (STAT3) axis [5,6]. EVs produced by lung cells in response to pro-inflammatory stimuli have been shown to contribute to a variety of pathologic responses [7,8,9]. On the other hand, our previous finding that resident alveolar macrophages (AMs)—the main immune cell in the distal lung—at steady-state tonically secrete EVs containing suppressor of cytokine signaling 3 (SOCS3) [10] exemplifies an opposing paradigm. Upon uptake of these EVs by neighboring alveolar epithelial cells (AECs) [11], the transcellular acquisition of vesicular SOCS3 serves as a critical restraint on cytokine-induced Janus kinase (JAK)-STAT3 signaling therein, which helps to maintain homeostasis in the distal lung [10,12,13]. We and others have demonstrated that secretion of vesicular constituents is a dynamically regulated phenomenon. For example, intrapulmonary administration of IL-10 augments, whereas lipopolysaccharide inhibits the secretion of SOCS3 in EVs by AMs [10]. Nevertheless, the specific cellular and molecular machinery that controls the packaging of cargoes of cytosolic origin into EVs represents a major gap in our understanding of EV biology. Furthermore, quantitative interrogation into how vesicular cargo packaging changes in response to exogenous and endogenous stimuli is currently lacking.

Catalytic inactivation of the proteasome by ROS is known to result in the accumulation of oxidized and polyubiquitinated proteins [14,15]. Proteasome inhibitors have emerged as an effective approach for anti-cancer therapy [16,17], and their efficacy has been attributed, at least in part, to modulation of the content and function of cancer cell-derived EVs that results in inhibition of processes such as angiogenesis [18]. Despite this knowledge, the influence of microenvironmental regulation of the proteasome on the packaging and secretion of EV cargoes has never been explored. Here, we demonstrate proteasomal control over vesicular secretion of SOCS3 by AMs and its dynamic alteration by a clinically relevant source of oxidative stress, cigarette smoke extract (CSE). We show that ROS augment AM release of SOCS3 in EVs by inactivating the proteasome. Furthermore, we describe a novel, semi-quantitative intracellular protein packaging assay that reveals specificity in the stimulatory effects of ROS and proteasome inhibition on the incorporation of SOCS3 into EVs. These studies advance both the mechanistic understanding of microenvironmental regulation of packaging and secretion of a vesicular cargo protein and the methodologic means for its evaluation.

## 2. Materials and Methods

### 2.1. Preparation of CSE

CSE was prepared by bubbling the smoke from 5 research cigarettes (Lot #3R4F, University of Kentucky Research Institute, Lexington, KY, USA) through 50 mL of RPMI 1640 culture medium (Thermo Fisher Scientific, Waltham, MA, USA) using a glass impinger (Ace Glass, Vineland, NJ, USA). CSE was then filtered through a 40 μm cell strainer (BD) and 1 mL aliquots of CSE were stored at −80°C [19].

### 2.2. Isolation and Culture of Primary and Immortalized AMs

Primary AMs were collected from lung lavage fluid of pathogen-free female Wistar rats (Charles River Laboratories, Wilmington, MA, USA), as described [20]. Animals were maintained at the University of Michigan Unit for Laboratory Animal Medicine, and studies were approved by the Institutional Animal Care and Use Committee. MH-S cells, an immortalized murine AM cell line (ATCC. Manassas, VA, USA), passaged 1:10 every 3–4 days, were grown in polystyrene flasks in RPMI 1640 culture medium supplemented with 10% FBS and penicillin/streptomycin. For treatment with oxidants and proteasome inhibitors, primary AMs and MH-S cells were plated in serum-free RPMI 1640 culture medium (1–1.5 × 10^6^ cells/mL) in 6-well, polystyrene plates. Cells were adhered for at least 1 h and then washed prior to treatment to remove EVs and other secreted products released during adherence to plastic. Cells were treated with CSE or H_2_O_2_ (Fisher Scientific) at the indicated concentrations for 1 h, washed, and then incubated in serum-free medium for the durations specified. For proteasome inhibition experiments, MH-S cells were treated with bortezomib or MG132 (both from MilliporeSigma, Burlington, MA, USA) at the indicated concentrations and for specified durations in serum-free medium. For experiments using antioxidants, cells were treated for 1 h with 50 μM of the broad-spectrum antioxidant *N*-acetyl-l-cysteine (NAC, Sigma-Aldrich, St. Louis, MO, USA) prior to stimulation with CSE. NAC was also present in the culture medium post-CSE stimulation to quench the production of endogenous ROS by cells.

### 2.3. EV Isolation

Conditioned medium (CM) from cells pulsed for 1 h with CSE or H_2_O_2_, or continuously treated with MG132 or bortezomib, was collected after 20 h culture. CM was sequentially centrifuged at 500× *g* for 10 min and 2500× *g* for 12 min to remove dead cells, cell debris, and apoptotic bodies. EVs were then isolated using two approaches [21]. (1) For rapid concentration of all EVs, CM was centrifuged at 4,000× *g* for 20 min in 100-kDa centrifugal filter units (MilliporeSigma), and the resulting > 100 kDa fraction was used to analyze secretion of EVs, SOCS3, and vacuolar protein sorting-associated protein 4a (VPS4a). (2) For fractionation of EVs by ultracentrifugation, CM was spun at 17,000× *g* for 160 min to pellet “large EVs (lEVs),” from which the supernatant (non-lEV fraction) was then spun at 100,000× *g* for 90 min to pellet “small EVs (sEVs)” [10]. The resulting lEV and sEV fractions were used for analysis of SOCS3 secretion.

### 2.4. Western Blot

For probing of cell lysates, protein concentrations were determined by DC protein assay (Bio-Rad, Hercules, CA, USA), and aliquots containing 10 μg protein were used for analysis. For probing of CM vesicular concentrate samples, entire > 100 kDa, lEV, or sEV fractions were collected from cells treated in culture and used to detect SOCS3. All samples were separated by SDS-PAGE using 12.5% gels and transferred to nitrocellulose membranes using Trans-Blot Turbo Mini Nitrocellulose Transfer Packs (Bio-Rad). Membranes were blocked for 1 h with 4% BSA and incubated overnight with commercially available monoclonal antibodies directed against SOCS3 (mouse, SO1, Abcam, Cambridge, GBR), VPS4a (rabbit, EPR14545(B), Abcam), or GAPDH (rabbit, 14C10, Cell Signaling Technology, Danvers, MA, USA). After washing and incubation with peroxidase-conjugated anti-mouse or anti-rabbit secondary antibodies, the film was developed using ECL detection (GE Healthcare, Chicago, IL, USA). Exposure times for each experiment were selected to optimize a wide linear dynamic range, ensuring detection of a control vesicular SOCS3 band while limiting—to the best of our abilities—saturation of enhanced vesicular SOCS3 bands resulting from treatment of AMs with ROS or proteasome inhibitors. Developed films were then scanned using a desktop scanner at a dots per inch of 300 or greater. The optical density (OD) for SOCS3 bands was quantified using NIH ImageJ software (Version 1.51, Bethesda, MD, USA) as an area under the profile curve. As consistently as possible, background noise was corrected for by enclosing each peak at the same distance from its baseline. When present as a double-banded signal in vesicular fraction ( > 100 kDa) samples, both SOCS3 bands were enclosed for OD quantification. Densitometry was expressed relative to the control values for each experiment.

### 2.5. ROS Assay

Oxidative stress in MH-S cells was determined using the well-established DCFDA/H2DCFDA Cellular ROS Assay Kit (Abcam) [22]. As previously described [23], cell-permeant DCFDA (also known as DCFH-DA) was added to cells where it became hydrolyzed by intracellular esterases to form non-fluorescent DCFH. In the presence of ROS, DCFH was oxidized to the fluorescent compound DCF, thus allowing for indiscriminate measurement of total ROS by quantifying fluorescence using a microplate reader.

To measure oxidants directly delivered to cells by CSE, adherent MH-S cells (2.5 × 10^4^) were labeled with DCFDA and then stimulated with CSE in 96-well, polystyrene plates. Fluorescence intensity (i.e., ROS) was determined after treatment for 1 h. To measure endogenous oxidants generated by cells in response to incubation with CSE, adherent MH-S cells were treated with CSE for 1 h, washed, and subsequently labeled with DCFDA. Fluorescence intensity was then determined 4 h post-treatment. To correct for background fluorescence contributed by particulates contained in CSE, fluorescence intensity from unlabeled MH-S cells was subtracted from values obtained for DCFDA-labeled cells treated with CSE in parallel. All data were expressed relative to control values for each experiment.

### 2.6. Nanoparticle Tracking Analysis (NTA)

The concentration and size distribution of EVs secreted by MH-S cells was determined using NanoSight NS300 (Malvern Panalytical, Malvern, GBR). Entire vesicular fraction (> 100 kDa) samples collected from CM were diluted to yield 10–80 particles per frame and injected via continuous infusion (flow rate ~15). Particles were analyzed over a 60 s capture period, and at least 3 capture periods were generated for each sample per experiment. Results indicating low concentration (< 10 particles per frame) and/or high vibration were omitted. Injection with at least 1 mL PBS was used to wash tubing in between samples.

### 2.7. Carboxyfluorescein Succinimidyl Ester (CFSE)-Based Vesicular SOCS3 Packaging Assay

To fluorescently label intracellular proteins in MH-S, cells were incubated in serum-free culture medium with 10 μM CFSE (Sigma-Aldrich) for 15 min at 37 °C. Excess CFSE was quenched by adding equal volumes of serum-containing culture medium, and cells were pelleted at 500× *g* for 10 min, adhered, and treated as stated above in 6-well, polystyrene plates. EVs were collected by 100-kDa filtration and vesicular CFSE (> 100 kDa) fluorescence was quantified for each experimental condition. SOCS3 was also probed by Western blot of >100 kDa vesicular fraction samples collected from unlabeled cells treated in parallel. Specific packaging of SOCS3 as compared to all intracellular proteins was then semi-quantitatively determined by dividing the densitometry value for vesicular SOCS3 (relative to control) by the raw CFSE fluorescence value quantified for parallel samples (vesicular SOCS3/vesicular CFSE).

### 2.8. 20S Proteasome Activity Assay

Measurement of 20S proteasome activity (relative to control) in MH-S cell lysates was determined using a 20S Proteasome Assay Kit (Cayman Chemical, Ann Arbor, MI, USA). Adherent cells (5 × 10^5^) were treated in 12-well, polystyrene plates and lysates were collected after 20 h culture. Fluorescence resulting from SUC-LLVY-AMC substrate proteolysis was used to calculate the catalytic activity of the proteasome in cells following treatment with CSE, H_2_O_2_, or bortezomib.

### 2.9. Data Collection and Analysis

Results were from at least 3 independent experiments containing single samples per condition unless specified otherwise in the figure legend. Pooled data were expressed as mean ± SEM and analyzed using the Prism 5.0 statistical program from GraphPad software. Significance was determined using a one-way ANOVA or paired student’s *t*-test, as appropriate, and was inferred at a *p* < 0.05. Significant values were labeled with varying numbers of asterisks (*), and corresponding *p* values were defined in the figure legends.

## 3. Results

### 3.1. CSE Enhances SOCS3 Secretion by Primary AMs in an ROS-Dependent Manner

Given the imperative to restrain inflammation in the oxygen-rich microenvironment of the lung, we hypothesized that AMs would respond to oxidative stress by increasing the release of vesicular SOCS3. To explore this, we used the well-established in vitro experimental tool of CSE, an aqueous extract of the common environmental toxin, cigarette smoke, which was known to both contain a myriad of exogenous oxidants [24,25,26] and to stimulate endogenous cellular production of ROS [27,28,29,30,31,32]. We exposed primary rat AMs with increasing doses of CSE for 1 h. Following treatment, AMs were washed and incubated for 20 h to allow for the elaboration of detectable levels of SOCS3-containing EVs into CM, as measured by Western blot. Treatment of primary AMs with CSE led to an increase in vesicular SOCS3 release, which was marked at a 7.5% dose (Figure 1A and Appendix A). This dose of CSE was within the range commonly employed in in vitro macrophage studies [33,34,35] and did not cause significant cytotoxicity in our experimental system (Appendix A). Additionally, CSE caused no significant change in the amount of intracellular SOCS3 present in AMs, though it was associated with some decrease in lysate GAPDH (Figure 1B and Appendix A). Given that decreased intracellular levels of GAPDH did not extend to SOCS3, along with our data showing minimal cytotoxicity in AMs, we speculated that this effect was attributable to either robust non-vesicular GAPDH release into CM, as had been previously described [3], and/or transcriptional/translational shutdown caused by oxidative stress. Together, these data suggested that CSE stimulated the release of intracellular SOCS3 from primary AMs, rather than by augmenting the intracellular pool of SOCS3 available for secretion.

To determine whether the increased secretion of SOCS3 was a consequence of the oxidant stress posed by CSE, we treated primary AMs with the general antioxidant NAC prior to CSE stimulation. NAC completely abrogated the stimulatory effects of CSE on vesicular SOCS3 secretion (Figure 1C and Appendix A). These data indicated that CSE potentiates SOCS3 secretion by primary AMs in a ROS-dependent manner.

### 3.2. Endogenous and Exogenous ROS Stimulate SOCS3 Release by Immortalized AMs

Given the relatively low yield of primary AMs that can be obtained by lung lavage, and that large numbers of AMs are required to detect a tonically secreted SOCS3 signal, we turned to MH-S cells—an immortalized line of cells derived by SV40 treatment of primary mouse AMs—to more thoroughly interrogate the mechanisms underlying ROS enhancement of vesicular SOCS3 secretion. To first determine whether immortalized AMs were appropriate for modeling the stimulatory effect of CSE on SOCS3 release observed in primary cells, we treated MH-S cells with increasing doses of CSE. Similar to the response observed for primary rat AMs, CSE elevated MH-S vesicular SOCS3 secretion in a threshold manner, an effect that became significant at a 3% dose (Figure 2A) and was unaccompanied by detectable levels of cytotoxicity (Appendix A). The increment in SOCS3 secretion exhibited by MH-S cells at 3% CSE was comparable to that displayed by primary AMs at 7.5%. Given the data in Figure 1C demonstrating a role for ROS, we speculated that this greater resistance to (i.e., a higher dose required for) the stimulatory effects of CSE on SOCS3 secretion in primary than immortalized AMs reflects their heightened antioxidant defenses acquired during residence in the oxygen-rich alveolar milieu.

To further investigate the oxidative stress response in MH-S cells, we first measured ROS levels following CSE treatment. Stimulation with CSE led to increases in MH-S cell ROS that paralleled those for SOCS3 secretion (Figure 2B). Furthermore, as with primary AMs, pre-treatment of MH-S cells with NAC completely abrogated the stimulatory effects of CSE on SOCS3 secretion (Figure 2C).

Accumulation of ROS in CSE-treated MH-S cells was not observed before 4 h following the addition of CSE (data not shown), suggesting that measured ROS were generated from endogenous cellular sources, rather than reflecting those delivered directly by the CSE itself. Although CSE has been reported to promote mitochondrial ROS production [27,28], pre-treatment of MH-S cells with the mitochondria-specific anti-oxidant MitoTEMPO (10 μM) had no effect on SOCS3 secretion in response to CSE (data not shown). Finally, to interrogate whether exogenous sources of ROS exerted the same effect as endogenous ROS on SOCS3 release, we treated MH-S cells for 1 h with increasing doses of the important oxidant species H_2_O_2_. In a similar manner to CSE, H_2_O_2_ treatment led to dose-dependent increases in SOCS3 secretion by MH-S cells (Figure 2D) without causing significant increases in cytotoxicity (Appendix A). In parallel, H_2_O_2_ also enhanced > 100 kDa release of VPS4a (Appendix A), an ATPase whose vesicular recruitment is required for endosomal sorting complexes required for transport (ESCRT)-dependent EV biogenesis [36], thus indicating active release of EVs. A similar but modest and inconsistent effect on vesicular VPS4a secretion was observed following the treatment of MH-S cells with CSE (data not shown). Taken together, these data demonstrate that both endogenous and exogenous sources of ROS significantly potentiate the active secretion of vesicular SOCS3 by MH-S cells.

### 3.3. CSE Augments MH-S Cell Secretion of SOCS3 by Stimulating Production of, and Packaging of SOCS3 into, EVs

We reasoned that the stimulatory effect of ROS on the secretion of vesicular SOCS3 could be caused by three mechanisms, either acting alone or in combination: 1) Increasing the intracellular pool of SOCS3 available for secretion in EVs; 2) enhancing the total number of EVs released; and/or 3) specifically augmenting the packaging of SOCS3 per released EV. To determine whether CSE increased intracellular expression of SOCS3, we measured protein levels in MH-S cell lysates by Western blot 20 h after exposure. As demonstrated for primary AMs (Figure 1B), CSE had no effect on SOCS3 expression in MH-S cells (Figure 3A). These results suggested that the stimulatory effects of ROS on SOCS3 secretion were independent of increased intracellular content, and instead reflect increased EV production and/or vesicular SOCS3 packaging.

As oxidative stress has been previously demonstrated to enhance generation of total EVs [7,37,38,39], we quantified EVs in MH-S cell CM using NTA. In accordance with these prior reports, elaboration of EVs from MH-S cells treated with CSE was increased (Figure 3B and Appendix A). Increases in EV numbers were localized to those ≥ 100 nm in diameter, which was consistent with the size range reported for lEVs [3,4]. To determine whether the stimulatory effect of CSE on SOCS3 secretion paralleled this effect on EV production, we performed differential ultracentrifugation using well-defined sedimentation rates [3,4] to isolate a 17,000× *g* lEV pellet and 100,000× *g* sEV pellet. As previously reported [10], basal SOCS3 secretion by MH-S cells was detected in lEVs but not sEVs (Figure 3C and Appendix A). Additionally, and consistent with our NTA data, CSE promoted robust increases in SOCS3 packaging in a lEV fraction but not sEV fraction (Figure 3C and Appendix A). These differential centrifugation data both confirmed the localization of SOCS3 in CM to EVs and further demonstrated that CSE potentiates vesicular release of SOCS3 specifically in lEVs.

The incremental increase in EV production (Figure 3B) was quantitatively insufficient to account for the relatively greater fold change in vesicular SOCS3 secretion caused by CSE (Figure 2A). These results suggested that, in addition to enhancing EV biogenesis—thus leading to non-specific increases in SOCS3 release—ROS may also augment the amount of SOCS3 specifically packaged per EV. As SOCS3 is a cytosolic protein, we developed a novel assay to semi-quantitatively assess the effects of CSE on vesicular release of SOCS3 relative to the total amount of all intracellular proteins packaged into MH-S cell-derived EVs. Specifically, we labeled MH-S cells with CFSE and quantified the relative fluorescence in a vesicular fraction following treatment with CSE as a measure of overall intracellular protein content in EVs. Consistent with the increased EV generation determined by NTA, treatment with CSE caused significant increases in the total amount of intracellular proteins (i.e., CFSE fluorescence) present in a vesicular fraction (Figure 3D). However, determination of the relative ratio of vesicular SOCS3 to vesicular intracellular protein content in parallel samples of CM from the same cultures revealed that CSE disproportionally augmented the release of SOCS3 in EVs (Figure 3D). These data suggested that some proportion of the oxidant-induced increment in intracellular protein packaging into MH-S-derived EVs was specific to SOCS3. To our knowledge, this semi-quantitative assay provided the most specific and unbiased interrogation of intracellular protein packaging yet reported. That CSE specifically enhanced SOCS3 packaging within EVs was further supported by employing an alternative but arguably less specific assay that determined the relative ratio of vesicular SOCS3 to the total amount of annexin V^+^ MVs (the subset of EVs previously reported [10] to preferentially encapsulate SOCS3) present in MH-S cell CM, as determined by Western blot and flow cytometry, respectively (Appendix A). In tandem, these data suggest that, in addition to increasing the total amount of EVs produced by MH-S cells, ROS also specifically augment the amount of SOCS3 packaged into secreted EVs.

### 3.4. Inhibition of the 20S Proteasome Mimics the Stimulatory Effects of ROS on Vesicular SOCS3 Secretion by MH-S Cells

Cargo sorting and biogenesis of plasma membrane (PM)-derived EVs are known to rely, at least in part, on the recruitment of adaptor proteins such as arrestin domain-containing protein 1 (ARRDC1) to the cell surface [40,41,42,43]. Furthermore, polyubiquitination of ARRDC1 is known to augment the production of PM-derived EVs [41]. The association observed between altered proteasome activity and SOCS3 secretion led us to hypothesize that ROS might stimulate SOCS3 packaging in MH-S cells by inhibiting the proteasome, thus potentially increasing the amount of polyubiquitinated ARRDC1 available to facilitate enhanced cargo sorting and EV biogenesis.

We first sought to confirm that ROS at the concentrations we employed inhibit catalytic activity of the 20S proteasome in MH-S cells. In accordance with prior observations in other cell types [44,45,46], treatment with H_2_O_2_ led to a dose-dependent reduction in 20S proteasome activity (Figure 4A); notably, the doses of H_2_O_2_ required for substantial inhibition of the 20S proteasome (100 μM and 250 μM) were those that most prominently increased vesicular release of SOCS3 by MH-S cells (Figure 2D). CSE also inactivated the 20S proteasome (Figure 4A), but only at the dose (3%) at which it augmented secretion of vesicular SOCS3 (Figure 2A). Pre-treatment of MH-S cells with NAC completely abrogated the inhibitory effect of CSE on 20S proteasome activity (Figure 4B), thus demonstrating that the ability of CSE to inhibit the 20S proteasome is ROS-dependent, just as is its ability to stimulate vesicular SOCS3 release. Analyzing all our experimental data with CSE and H_2_O_2_, there was a strong quantitative correlation between the degree of proteasome inhibition within MH-S cells and the amount of SOCS3 secretion within EVs elicited by ROS (Figure 4C).

Although these data established a strong association between oxidant enhancement of SOCS3 secretion in EVs and inhibition of 20S proteasome activity in MH-S cells, we sought to definitively demonstrate a causal role for the proteasome in controlling vesicular SOCS3 secretion. To do so, we treated MH-S cells with two known proteasome inhibitors, MG132 and bortezomib, hypothesizing that proteasome inactivation by pharmacologic agents would mimic the proteasome-inhibitory and SOCS3 secretion-stimulatory effects of ROS. Indeed, MG132 substantially augmented levels of SOCS3 released by MH-S cells (Figure 5A). Likewise, bortezomib, a highly potent inhibitor of 20S proteasome activity (Appendix A) [47], also augmented vesicular SOCS3 release by MH-S cells (Figure 5B), albeit at 10-fold lower concentrations than MG132; this effect was independent of any apparent cellular cytotoxicity (Appendix A) and involved parallel increases in the release of vesicular VPS4a (Appendix A). Taken together, these results are suggestive of a model in which at baseline, the proteasome acts as an endogenous brake on MH-S cell secretion of SOCS3, which can be disrupted by ROS to actively potentiate SOCS3 release in EVs.

### 3.5. Proteasome Inactivation in MH-S Cells Augments SOCS3 Secretion by Stimulating Production of, and Packaging of SOCS3 into, EVs

As with CSE exposure, we next sought to determine whether the stimulatory effect of proteasome inhibitors on SOCS3 secretion by MH-S cells reflected increases in intracellular SOCS3 expression, production of EVs, and/or vesicular packaging of SOCS3. As our data suggested that oxidative inactivation of the proteasome was responsible for the increases in SOCS3 secretion promoted by CSE (Figure 4 and Figure 5), we hypothesized that proteasome inhibitors similarly acted by increasing both EV production and SOCS3 packaging. Indeed, treatment of MH-S cells with either MG132 or bortezomib failed to significantly alter intracellular SOCS3 expression (Figure 6A). Although bortezomib at a dose of 1 μM caused increased production of EVs with diameters < 100 nm and ≥ 100 nm (Figure 6B and Appendix A), its stimulatory effect on SOCS3 secretion was, as with that of CSE, localized to a 17,000× *g* lEV pellet but not a 100,000× *g* sEV pellet (Figure 6C and Appendix A). Furthermore, we noted that the <two-fold change in EV production was insufficient to account for the ~30-fold increase in vesicular SOCS3 secretion caused by bortezomib (Figure 5B). To semi-quantitatively determine if bortezomib, like ROS, promoted increases in vesicular SOCS3 packaging, we employed the CFSE-based SOCS3 packaging assay to show that bortezomib disproportionally augmented release of SOCS3 in EVs relative to other intracellular proteins (Figure 6D). Such an effect on packaging was also supported by determining the relative ratio of vesicular SOCS3 to the total amount of annexin V^+^ MVs present in MH-S cell CM (Appendix A). Therefore, taken together, our data demonstrate that proteasome inhibitors act in the same manner as CSE to both elevate production of EVs and to specifically augment the amount of SOCS3 packaged into secreted EVs.

## 4. Discussion

Using CSE treatment as a clinically relevant in vitro model for oxidative stress within the alveolar microenvironment, we demonstrate that ROS augment vesicular SOCS3 release by primary and immortalized AMs (i.e., MH-S cells), a phenomenon attributable at least in part to catalytic inactivation of the 20S proteasome. These results are the first to suggest that proteasomal control over secretion of EVs and vesicular cargoes is subject to dynamic regulation by microenvironmental stimuli. Further, our findings expand on the limited body of literature illuminating the critical importance of proteasome activity for dictating extracellular elaboration of vesicular content [18]. Although we anticipate that this interplay between microenvironmental signals, proteasome function, and EV number/composition extends to other tissue contexts, the importance of oxidative stress as a determinant of these processes may be especially meaningful in the oxidant-rich environment of the distal lung. Although the incremental increases in MH-S cell vesicular SOCS3 secretion caused by ROS and proteasome inhibitors are quantitatively modest, our previous findings predict that changes of this magnitude are sufficient to significantly dampen STAT3 signaling in AECs [12]. We, therefore, anticipate that augmented SOCS3 release by AMs assumes an important role for inflammatory restraint in the hyperoxic alveolar milieu.

We also sought to mechanistically determine whether the alterations in SOCS3 release by MH-S cells subjected to oxidant stress and proteasome inhibition involved changes in intracellular SOCS3 expression, EV secretion, and/or specific packaging of SOCS3 into EVs. To our knowledge, no previous study has applied such semi-quantitative methods to interrogate the mechanistic basis for regulated packaging of a vesicular cargo. We developed a novel CFSE-based packaging assay to semi-quantitatively assess SOCS3 release relative to other EV-packaged intracellular proteins, all of which would be expected to be labeled to a comparable extent by CFSE. We consider this unbiased and global readout of intracellular protein secretion to be superior to analyses based on measuring the secretion of a single such protein (e.g., β-actin) [48], which is likely to be subject to its own dynamic regulation by endogenous/exogenous stimuli. Using this assay, we found that the stimulatory effects of CSE and bortezomib on SOCS3 release were at least in part a reflection of its greater specific packaging within MH-S cell-derived EVs. However, we cannot fully discount the possibility that passive incorporation of cytosolic content into budding EVs contributed to observed increases in SOCS3 secretion. Future studies should couple packaging assays with biochemical characterization of a vesicular fraction to additionally confirm absence of non-vesicular cytosolic constituents in the context of an exogenous stimulus. Nevertheless, our findings advance semi-quantitative assessment of intracellular protein packaging and highlight the need to develop analogous modalities for measuring the sorting of vesicular cargoes originating in other cellular compartments.

We acknowledge three methodologic concerns that limit our overall stated conclusions. The first of these is our use of Western blot to semi-quantitatively assess changes in vesicular SOCS3 release. Despite our best efforts to correct for background and optimize a linear dynamic range for each experiment, it is nonetheless likely that the densitometry value measured for vesicular SOCS3 by Western blot is not a perfect linear correlate to its absolute concentration. Therefore, although the semi-quantitative effects of ROS and proteasome inhibitors on SOCS3 packaging and secretion were strikingly consistent across experiments, the exact magnitude of these effects remain unclear. It will be important for follow-up studies to employ more precise methodologies that enable the measurement of absolute changes in EV content to fully appreciate the influence of microenvironmental factors on regulated cargo sorting. Second is the limitation of using 100-kDa filtration for measuring vesicular SOCS3 and vesicular CFSE fluorescence given the myriad non-EV mechanisms that could contribute to the release of cytosolic content into CM. Cytotoxicity assays revealed no accumulation of extracellular LDH following treatment of MH-S cells with CSE or bortezomib, thus arguing against a meaningful contribution of cell lysis to the amount of SOCS3 and CFSE measured in the > 100 kDa concentrate. Additionally, given our foundational report describing an unconventional, temperature-dependent mechanism of SOCS3 secretion specifically confined to annexin V^+^ lEVs pelleted at 17,000× *g*, here we used ultracentrifugation to confirm localization of SOCS3 in CM to lEVs and verify that changes in SOCS3 release detected by 100 kDa size-exclusion paralleled those observed in a 17,000× *g* pellet [10]. Data showing comparable changes in SOCS3 secretion using these two approaches support our conclusion that the effects of oxidative stress and proteasome inhibition on SOCS3 release are specific to a vesicular fraction. As further evidence that ROS and proteasome inhibitors actively potentiate SOCS3 secretion within EVs rather than via some passive, non-specific mechanism, we observed increased vesicular (i.e., > 100 kDa) secretion of VPS4a by MH-S cells treated with ROS and bortezomib. As vesicular recruitment of VPS4a is required for EV biogenesis [36], our results—which paralleled the effect of these treatments on SOCS3 release by MH-S cells—strongly suggest that oxidative stress/proteasome inactivation specifically and actively promote EV cargo sorting, biogenesis, and release. Additionally, we observed near parallel increases in EV number and vesicular CFSE fluorescence caused by CSE and bortezomib using our novel packaging assay. These data argue in favor of EVs as being the predominant contributor to CFSE fluorescence detected in a > 100 kDa concentrate and support our conclusion of enhanced vesicular SOCS3 sorting relative to the total intracellular protein pool. Lastly, we acknowledge the limitation of defining EV fractions according to sedimentation velocity without providing physical or biochemical verification of EV size, purity, or enrichment of known molecular markers. Importantly, we observed parallel increases in the secretion of ≥ 100 nm EVs (by NTA) and packaging of SOCS3 in a 17,000× *g* pellet following treatment of MH-S cells with CSE and bortezomib. Importantly, these treatments had minimal or no effect on secretion of < 100 nm EVs and packaging of SOCS3 in a 100,000× *g* pellet. Therefore, our results strongly argue in favor of the conclusion that ROS and proteasome inhibitors achieve their stimulatory effects on SOCS3 secretion by specifically augmenting packaging in lEVs. However, it will be important for future studies to demonstrate parallel enrichment of known markers—for example, α-actinin-4 in lEVs and CD81 in sEVs [3,4]—to corroborate specificity in EV cargo sorting.

ROS effects on miRNA and mRNA content of EVs have been previously described [7,39,49,50], though data on ROS regulation of vesicular protein sorting remains limited [51]. To our knowledge, oxidative stress has not been demonstrated to specifically alter the packaging of a protein resident to the cytosol. The extant body of literature reporting ROS effects on EVs is limited to studies of the effect of exogenous H_2_O_2_. Our data additionally demonstrated that the generation of endogenous ROS via treatment of MH-S cells with CSE was sufficient for augmenting SOCS3 release. CSE is known to promote the production of intracellular ROS primarily through its actions on nicotinamide adenine dinucleotide phosphate (NADPH) oxidase and mitochondrial activity [27,28,29,30,31,32]. Our results suggested that the effects of CSE on stimulated SOCS3 release by MH-S cells did not involve mitochondrial ROS generation. Therefore, we suspect that the activation of NADPH oxidase complexes by CSE caused elevated levels of ROS and corresponding increases in SOCS3 secretion by AMs. However, the involvement of other enzymatic (e.g., lipoxygenase, xanthine oxidase, etc.) and organellar (e.g., endoplasmic reticulum) sources of ROS following exposure to CSE [52,53,54,55,56] is possible. It will be informative in future studies to determine whether all sources of intracellular ROS act similarly to alter mechanisms of EV biogenesis and/or cargo packaging, or whether there is specificity in modulatory actions for ROS generated by particular enzymes and/or in particular subcellular compartments.

Despite the critical importance of protein turnover for maintaining cellular homeostasis and the recognition that proteasome components are packaged in EVs [57,58,59,60], investigation into the functional crosstalk between proteasome activity and EVs is limited. Bortezomib was first reported to suppress the production of EVs by T-lymphocytes activated to undergo apoptosis by UVB irradiation [57]. These results suggested that the proteasome promotes the secretion of EVs in the presence of an exogenous stimulus. However, a recent report using resting multiple myeloma cells demonstrated a stimulatory role for bortezomib on production of EVs [18]. These data highlight differences in the directionality of proteasome regulation of EV biogenesis, depending on whether an activating stimulus is present. Our study using MH-S cells supports a model in which—at steady state—the proteasome acts as a brake on vesicular SOCS3 packaging and EV secretion, which can be released upon its inactivation by oxidative stress. While our quantitative packaging data demonstrate augmented sorting of SOCS3 into EVs by exposure of MH-S cells to CSE and bortezomib, the effects of these stimuli on the packaging of global cargo contents of AM-derived EVs is unknown. Notably, Zarfati et al. demonstrated broad alterations in pro-inflammatory and pro-angiogenic growth factor proteins present in EVs elaborated from multiple myeloma cells following bortezomib treatment. Given the growing interest in the use of bortezomib as a therapeutic, not merely in the context of multiple myeloma [61] but also in tissue fibrosis [62,63,64], it is intriguing to speculate that the drug’s protective effects are in part mediated by changes in vesicular communication, as has been previously suggested [18].

Although increases in vesicular SOCS3 release and proteasome inhibition during oxidative stress were strongly correlated (Figure 4), we were limited in our ability to causally link these two processes. Our attempts to overcome the stimulatory effects of ROS on SOCS3 release by overexpressing proteasome catalytic subunits were thwarted by low transfection efficiency. Additionally, treatment of MH-S cells using small molecule proteasome activator compounds was ineffective due to the non-specific nature of these drugs. Consequently, we cannot exclude the possibility that some proportion of the augmentation of SOCS3 secretion by ROS is independent of proteasome inactivation. Given the breadth of ROS effects on cellular function, it is likely that regulation of vesicular cargo release by oxidants is mechanistically multifaceted. Nonetheless, taken together, our data do argue that proteasome inhibition represents at least one important mechanism by which oxidative stress potentiates the packaging and release of cytosolically packaged vesicular proteins.

While our foundational report described tonic SOCS3 secretion exclusively in annexin V^+^ MVs pelleted at 17,000× *g* [10], it is still unclear whether stimulated release by ROS and/or proteasome inhibitors involved SOCS3 packaging and release within both MVs and Exos. Of note, our NTA data indicated that the majority of CSE effects and much of bortezomib effects on stimulated EV release by MH-S cells were specific to those with a diameter of 100–300 nm, consistent with the size range reported for lEVs, including classical MVs [1]. However, bortezomib did also cause a notable increase in EVs < 100 nm in diameter, thus raising the possibility that SOCS3 undergoes stimulus-specific packaging into smaller MVs or Exos. Nevertheless, leveraging the well-defined sedimentation properties of EVs [3,4] we show that the stimulatory effects of CSE and bortezomib on SOCS3 secretion were entirely exclusive to a 17,000× *g* lEV pellet but not 100,000× *g* sEV pellet collected from MH-S cells. Therefore, we suggest that proteasome inhibition causes heightened packaging and release of SOCS3 predominantly within larger PM-derived MVs, but cannot discount the possibility that SOCS3 is subject to some amount of vesicular sorting via an endosomal pathway. Accordingly, given the potential crosstalk between secretory endocytic and autophagic systems [65], we also cannot eliminate a potential role for autophagy in elaborating an unconventional mechanism of vesicular SOCS3 release. Indeed, ROS [66,67,68] and proteasome inhibitors [69,70] are known to enhance autophagy, thus raising the possibility that these stimuli might potentiate SOCS3 release by AMs by increasing fusion and secretion of autophagic/endosomal compartments. We are currently developing a fluorescent SOCS3 conjugate to mechanistically delineate its packaging in AMs.

The molecular adaptors and targeting elements that are responsible for vesicular sorting of biomolecules, particularly for cytosolic proteins packaged at the PM, represent a major gap in current knowledge [71]. However, it is known that, in addition to serving as the post-translational signal responsible for targeting of substrates to the proteasome, polyubiquitination of the cargo-sorting adaptor protein ARRDC1 also enhances biogenesis of MVs at the PM [40,41,42,43]. This is achieved via ARRDC1 recruitment of ESCRT machinery, which is critical for vesicle budding and release [36,41,72]. Accumulation of intracellular polyubiquitinated substrates is a known consequence of proteasome inhibition [14,15]. Therefore, we speculate that oxidative inactivation of the proteasome potentiates SOCS3 release from AMs due to increases in polyubiquitinated ARRDC1, thus causing enhanced cargo sorting and MV secretion from the PM. Determining the role of ARRDC1 in SOCS3 packaging, both at steady-state and under stimulatory conditions, is of substantial interest to our laboratory; unfortunately, our preliminary efforts to evaluate this possibility were hindered by the lack of reagents applicable to non-human experimental systems.

In summary, we have illuminated a previously unknown mechanism in which proteasomal control over vesicular secretion of the cytosolically localized protein SOCS3 is tuned in AMs by oxidative stimuli. We have also developed a novel assay to demonstrate that the enhanced release of vesicular SOCS3 caused by oxidative inactivation of the proteasome is at least in part due to specific increases in vesicular SOCS3 packaging. Future studies will focus on identifying the molecular chaperones and/or targeting elements sensitive to proteasome inhibition and responsible for the sorting of SOCS3 into EVs. Further, follow-up experimentation will determine the breadth of proteasome effects on cargo content within, and on functional roles of, AM-derived EVs in the alveolar microenvironment.

## Figures and Tables

**Figure 1 cells-09-01589-f001:**
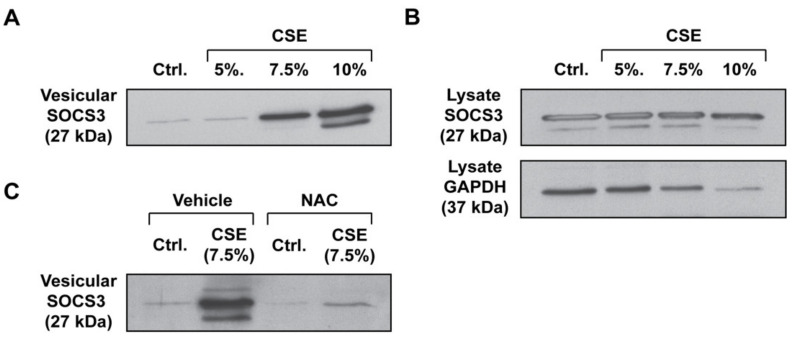
CSE stimulates the secretion of vesicular SOCS3 by primary AMs in a ROS-dependent manner. (**A**–**C**), Adherent AMs collected by lung lavage were treated with specified concentrations of CSE for 1 h. Following CSE treatment, AMs were washed and incubated for 20 h. (**A**) EVs in CM were concentrated by 100-kDa centrifugal filtration and vesicular SOCS3 secretion was determined via Western blot of the total vesicular fraction (>100 kDa) samples. Data are from 1 experiment representative of 3 independent experiments. (**B**) Following CSE stimulation (1 h) and incubation of AMs for 20 h, lysates were collected and subjected to Western blot for determination of SOCS3 expression, with GAPDH as a loading control. Data are from 1 experiment representative of 3 independent experiments. (**C**), Prior to CSE (7.5%) stimulation of AMs, NAC (50 μM) was added to AMs and maintained for the subsequent 20 h incubation. SOCS3 in the EV fraction of CM was analyzed by Western blot. Data are from 1 experiment representative of 3 independent experiments. Ctrl. = control.

**Figure 2 cells-09-01589-f002:**
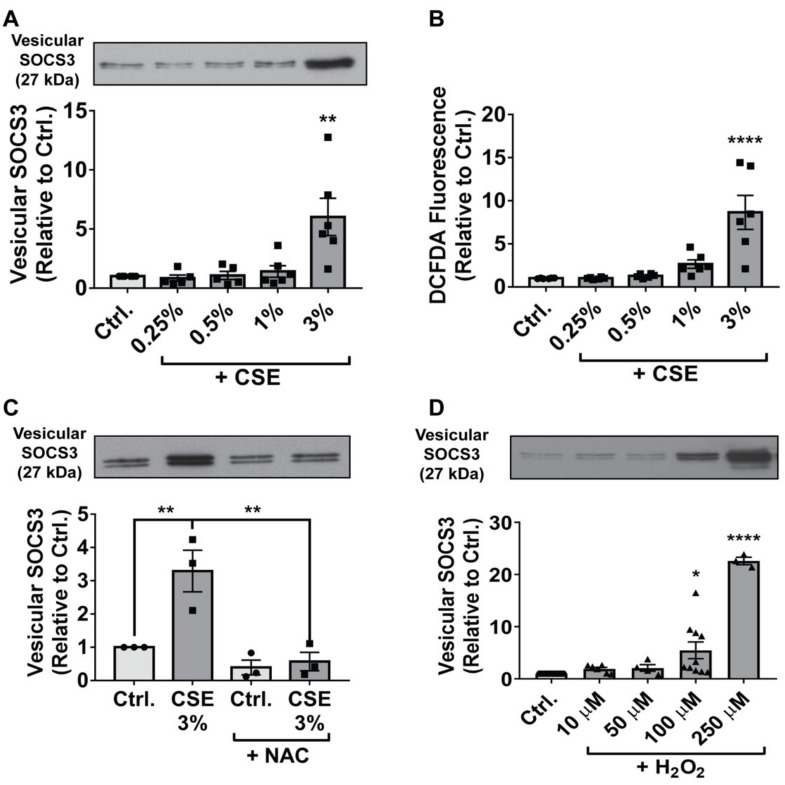
ROS augment vesicular SOCS3 secretion by immortalized AMs. (**A**) Adherent MH-S cells were treated for 1 h with indicated concentrations of CSE. Following CSE stimulation, cells were washed and incubated for 20 h. EVs were concentrated by 100-kDa centrifugal filtration, and SOCS3 secretion was determined by Western blot of total vesicular fraction (>100 kDa) samples. Densitometry data (mean ± SEM) are from >3 independent experiments, and significance analyzed by one-way ANOVA. (**B**) Adherent MH-S cells were treated with specified concentrations of CSE for 1 h, washed, and labeled with DCFDA. Fluorescence intensity was measured after 4 h culture. Duplicate samples were analyzed for each condition, and data (mean ± SEM) are from 3 independent experiments. Significance was analyzed by one-way ANOVA. (**C**) Prior to CSE (3%) treatment of MH-S cells, NAC (50 μM) was added for 1 h and maintained for the subsequent 20 h incubation. SOCS3 secretion was analyzed by Western blot of vesicular fraction samples. Densitometry data (mean ± SEM) are from 3 independent experiments, and significance analyzed by one-way ANOVA. (**D**) Adherent MH-S cells were treated with specified concentrations of H_2_O_2_ for 1 h. Following treatment, cells were washed and cultured for 20 h. Vesicular fraction samples were harvested and probed for SOCS3 by Western blot. Densitometry data (mean ± SEM) are from >3 independent experiments, and significance analyzed by one-way ANOVA. Ctrl. = control. *, ** and ****, *p* < 0.05, *p* < 0.01 and *p* < 0.0001, respectively vs. control. “Circles” indicate data points for Ctrl. samples, “squares” indicate data points for CSE samples, and “triangles” indicate data points for H_2_O_2_ samples.

**Figure 3 cells-09-01589-f003:**
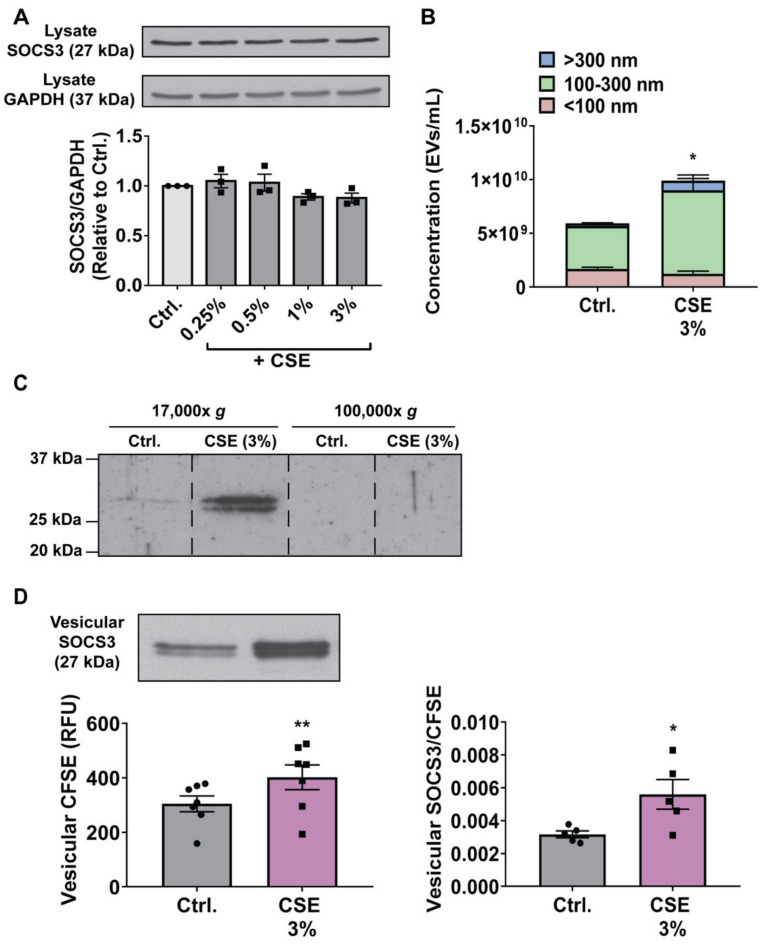
CSE potentiates MH-S cell production of EVs and packaging of vesicular SOCS3. **(A**–**B**) Adherent MH-S cells were treated (1 h) with specified concentrations of CSE, washed, and cultured for 20 h. (**A**) Lysates were collected and SOCS3 expression was assessed by Western blot, with GAPDH as a loading control. Densitometry data (mean ± SEM) are from ≥3 independent experiments, and significance analyzed by one-way ANOVA. (**B**) EVs were collected by 100-kDa centrifugal filtration, diluted in PBS, and quantified by NTA. Data are from 3 independent experiments in which ≥3 capture periods (60 s) were analyzed for each sample. Significance was determined using a paired sample *t*-test. (**C**) EVs were isolated by sequential ultracentrifugation of CM at 17,000× *g* and 100,000× *g* to pellet lEVs and sEVs, respectively, and probed for SOCS3 via Western blot. Dashed lines indicate splicing of discontinuous lanes from the same blot. Data are from 1 experiment representative of 3 independent experiments. (**D**) Prior to treatment (1 h) with CSE, MH-S cells were labeled with CFSE. Unlabeled cells were also treated with CSE in parallel for measuring secretion of vesicular SOCS3 into CM. Following treatment, wash, and culture of cells for 20 h, CM was collected, and vesicular fraction samples were either probed for SOCS3 via Western blot (*left panel*, *top*) or CFSE fluorescence determined (*left panel, bottom*). SOCS3 packaging was then measured, as described in “Experimental procedures” (*right panel*). Data (mean ± SEM) are from >3 independent experiments, and significance was determined using paired sample *t*-test. Ctrl. = control. * and **, *p* < 0.05 and *p* < 0.01, respectively vs. control. “Circles” indicate data points for Ctrl. samples and “squares” indicate data points for CSE samples.

**Figure 4 cells-09-01589-f004:**
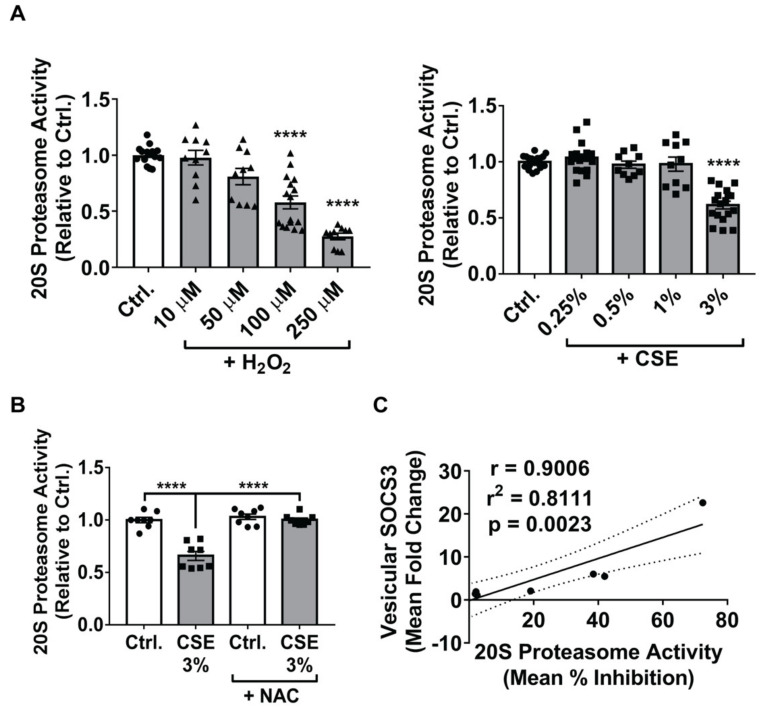
Oxidative inactivation of the proteasome in MH-S cells coincides with enhanced release of vesicular SOCS3. (**A**–**B**) Adherent MH-S cells were treated with indicated concentrations of H_2_O_2_ or CSE for 1 h, washed, and cultured for 20 h. Lysates were collected and proteasome activity was determined. Duplicate samples were analyzed for each condition, and data (mean ± SEM) are from >3 independent experiments. Significance was determined by one-way ANOVA. (**B**) Prior to CSE (3%) stimulation of MH-S cells, NAC (50 μM) was added to cells for 1 h and maintained for the subsequent 20 h incubation. Duplicate samples were analyzed for each condition, and data (mean ± SEM) are from >3 independent experiments. Significance was determined by one-way ANOVA. (**C**) Vesicular SOCS3 secretion data collected for every dose of CSE and H_2_O_2_ specified in Figure 2A,D were correlated with 20S proteasome activity data collected for parallel CSE and H_2_O_2_ treatments, as specified in Figure 4A. Results are depicted on an XY plot with the correlation coefficient (r and r^2^) and *p*-value (two-tailed) stated. Ctrl. = control. ****, *p* < 0.0001 vs. control. In (**A**–**B**) “circles” indicate data points for Ctrl. samples, “triangles” indicate data points for H_2_O_2_ samples, and “squares” indicate data points for CSE samples.

**Figure 5 cells-09-01589-f005:**
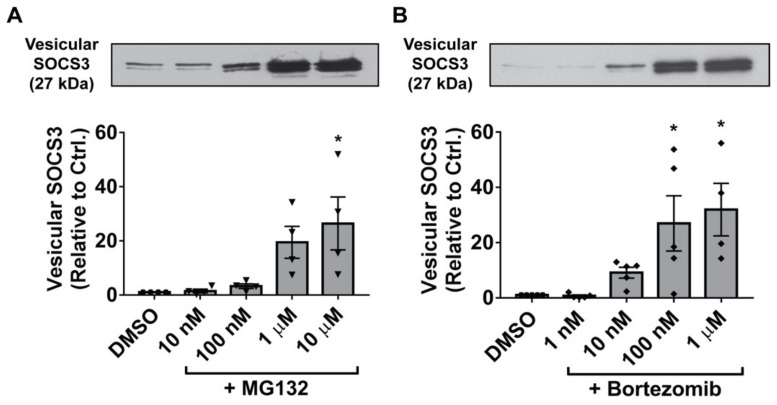
Inhibition of the proteasome stimulates MH-S cell secretion of vesicular SOCS3. (**A**–**B**) Adherent MH-S cells were treated with specified concentrations of MG132 or bortezomib for 20 h. EVs were harvested by 100-kDa centrifugal filtration and SOCS3 secretion was determined by Western blot of total vesicular fraction (>100 kDa) samples. Densitometry data (mean ± SEM) are from >3 independent experiments, and significance analyzed by one-way ANOVA. DMSO = DMSO control. *, *p* < 0.05 vs. DMSO control. “Hexagons” indicate data points for DMSO samples, “inverted triangles” indicate data points for MG132 samples, and “diamonds” indicate data points for bortezomib samples.

**Figure 6 cells-09-01589-f006:**
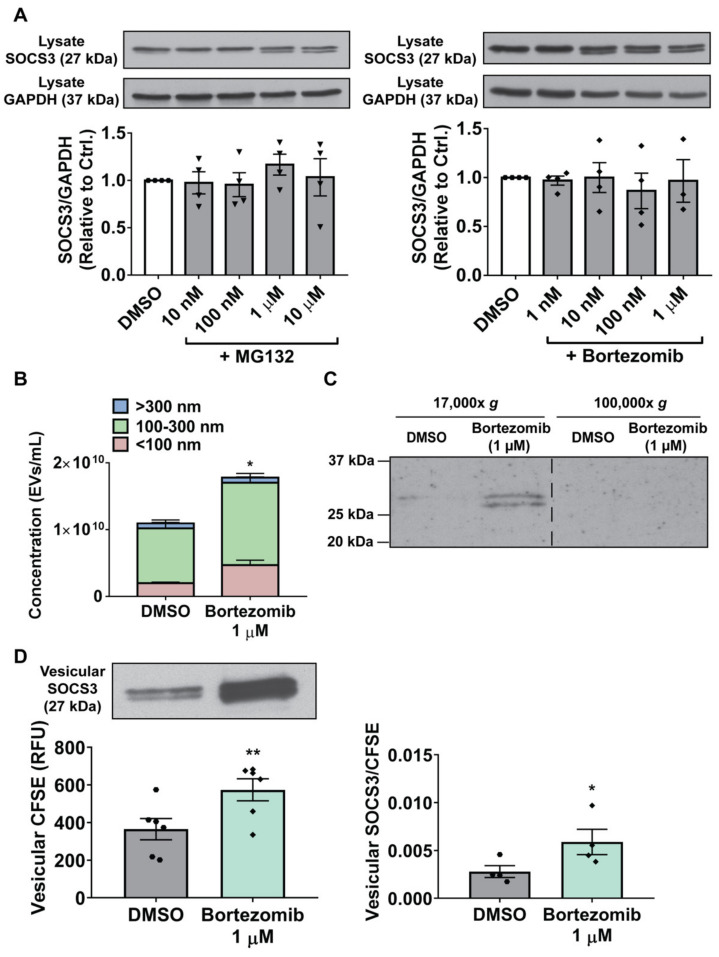
Inactivation of the proteasome potentiates MH-S cell production of EVs and packaging of vesicular SOCS3. (**A**–**B**) Adherent MH-S cells were treated for 20 h with specified concentrations of MG132 or bortezomib. (**A**) Lysates were collected and SOCS3 expression was assessed by Western blot, with GAPDH as a loading control. Densitometry data (mean ± SEM) are from ≥3 independent experiments, and significance analyzed by one-way ANOVA. (**B**) EVs were collected by 100-kDa centrifugal filtration, diluted in PBS, and quantified by NTA. Data (mean ± SEM) are from >3 independent experiments in which ≥3 capture periods (60 s) were analyzed for each sample. Significance was determined using paired sample *t*-test. (**C**) EVs were isolated by sequential ultracentrifugation of CM at 17,000× *g* and 100,000× *g* to pellet lEVs and sEVs, respectively, and probed for SOCS3 via Western blot. Dashed line indicates splicing of discontinuous lanes from the same blot. Data are from 1 experiment representative of 3 independent experiments. (**D**) Prior to stimulation with bortezomib, MH-S cells were labeled with CFSE. Unlabeled cells were also treated with bortezomib in parallel for measuring secretion of vesicular SOCS3. Following treatment, wash, and culture of cells for 20 h, vesicular fraction samples were concentrated via 100-kDa centrifugal filtration and either probed for SOCS3 via Western blot (*left panel*, *top*) or CFSE fluorescence (*left panel, bottom*). SOCS3 packaging was then measured, as described in “Experimental procedures” (*right panel*). Data (mean ± SEM) are from >3 independent experiments, and significance analyzed using paired sample *t*-test. DMSO = DMSO control. * and **, *p* < 0.05 and *p* < 0.01, respectively vs. DMSO control. “Hexagons” indicate data points for DMSO samples, “inverted triangles” indicate data points for MG132 samples, and “diamonds” indicate data points for bortezomib samples.

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
