# Peer review of "Oxidative Inactivation of the Proteasome Augments Alveolar Macrophage Secretion of Vesicular SOCS3"

_cells, 2020, doi:10.3390/cells9071589_

Round 1

Reviewer 1 Report

The authors present a detailed study of the effects of cigarette smoke and oxidative stress upon the release of extracellular vesicles from alveolar macrophages. The rationale is well explained and thorough descriptions are provided of the vesicle preparation and other techniques used, with one exception discussed below. The limitations are generally acknowledged, for example the choice of a 100kDa filter is discussed.

The data provide convincing evidence that CSE/oxidative stress causes an increase in the secretion of EVs and the SOCS3 associated with them. However, more thorough quantitation of SOCS3 expression, or at the very least a much more detailed explanation of the steps taken to ensure quantitative data, is required to support the assertion that ‘The incremental increase in EV production (Figure 3B) was quantitatively insufficient to account for the relatively greater fold change in vesicular SOCS3 secretion caused by CSE (Figure 2A).’

The CFSE packaging assay described is an ingenious approach to assess the relative contribution of a specific protein to overall EV cytosolic protein cargo. The authors explain that ‘Specific packaging of SOCS3 as compared to all cytosolic proteins was then determined by dividing the densitometry value for vesicular SOCS3 (relative to control) by the raw CFSE fluorescence value quantified for parallel samples (vesicular SOCS3/vesicular CFSE)’. However, the quantification of SOCS3 protein from Western blots (itself a ratio) as currently described is too unreliable to conclude that the observed changes in SOCS3/CFSE ratio with CSE stimulation are significant.  

The use of  ECL detection with film and subsequent densitometry as even a semi-quantitative measure is fraught with difficulties. The relationship between the concentration of each protein being assessed and the densitometry value is almost certainly non-linear and would need to be assessed across the range of protein concentrations observed. Comparison with a reference protein makes the situation more complicated and introduces further variability. The ECL signal is not constant and exposure times must be precisely recorded.  The guidelines suggested in the many publications warning of the difficulties of this approach should be followed (eg. Gassmann et al. Electrophoresis 2009, 30, 1845–1855; Butler TAJ et al Biomed Res Int. 2019; 2019: 5214821). The procedure used for the ‘quantification’ of Western blots needs to be described in much more detail, eg how was area of interest determined, what background correction was used, what dilution series was used for validation etc.

The data presented clearly indicates an increase in SOCS3 and there is convincing evidence that CSE increases the number of EVs released (eg Fig 3B) – indeed there are more larger EVs which could mediate a disproportionate increase in SOCS3. However, a more thorough quantification of SOCS3 is required to dissect the contribution of an increase in EVs and selective packaging of SOCS3. One approach might be to quantify CFSE and SOCS3 in the same number of vesicles from control and CSE stimulated cells. This would remove the need for a reference and if CFSE signal remained ~constant and SOCS3 increased would be convincing evidence of selective packaging.

In summary, this is a well-performed study presenting novel findings about the effects of CSE on alveolar macrophage EV production and  mechanism. However, more quantitative methods are required to support the most exciting conclusion that regulated packaging of the vesicular cargo SOCS3 is occurring.

Minor points

Replicate experiments and full size original images of Western blots, eg for Fig 1, should be made available in supplementary materials (or at least to reviewers).

Comment on the potential cytotoxicity of 7.5% CSE observed in Supplemental Fig 1. It may not reach significance p<0.05 in the test used, but 7.5% and 10% seem to be having more effect than in control.

Fig 1A. Clarify purpose of GAPDH. In later figs state that use ‘GAPDH as a loading control’. If it is an ‘invariant’ reference does this not suggest that SOCS3 in the lysate increases? It would be helpful to  employ additional reference proteins.

Why show representative images in Fig1 and densitometry data in Fig 2?

In Fig 2 the values plotted for the control are all 1. How is variance in the control readings included in the assessment of significance of differences?

Reviewer 2 Report

In this manuscript, Mikel et al examine the roles of oxidative stress on the SOCS3 EVs packaging and secretion by inactivation of the proteasome with an in vitro model by using Cigarette smoke extract (CSE) exposure of primary AMs and cell line. This is a very interesting study to be published with some revision.

Particularly, it will be much better if controls for the vesicular fraction were provided on the western blot data for SOCS3.  Any control will be helpful. I am wondering if oxidative stress specifically promotes SOCS3 packaging into EVs and secretion or in another way to stimulates the EVs formation and secretion non-specifically.

Reviewer 3 Report

The present study by Haggadone et al. investigates the role of oxidative stress in the unconventional release of cytoplasmic suppressor of cytokine signaling 3 (SOCS3) by alveolar macrophages (AM). This work is incremental and based on previous findings by the same group (PMID: 25847945) in which they proposed that AM-derived extracellular vesicles (EVs) containing SOCS1 and SOCS3, modulate inflammatory signaling in EV-recipient cells.

Now, the authors hypothesized that AMs may release high levels of SOCS3-containing EVs to blunt the inflammatory response elicited by high oxidative stress present in the alveolar region.

Unconventional release of “leaderless cargoes” (i.e. IL-1beta) into the extracellular space is an important topic with relevance in a number of research areas, thus the present study could be of interest to a broad range of readers. Nevertheless, before publication is recommended, the authors need to read carefully and address my comments with regards to both conceptual and technical flaws present in the current manuscript:

1) In this study, the authors propose that large EVs are responsible for releasing SOCS3.

There are no typical EV markers (CD81, CD63, CD9 etc.) shown in any of the immunoblots presented throughout the paper. Moreover, the authors should biochemically characterize better the EV fractions they are isolating, perhaps by blotting for Mitofilin or GP96; two proteins that were previously found to be enriched in large EVs (PMID: 26858453). Finally, a cellular fraction marker (i.e. ER, Tubulin) should also be shown to be excluded from purified EV fractions.

2) In order to gain insight into the mechanisms that regulate EV-mediated release of SOCS3, I would like to see whether siRNA depletion of ESCRT subunits (i.e. TSG101, VPS4) and/or Rab27, affects SOCS3 secretion. The authors can also treat cells with GW4869 to investigate the role of a ceramide-dependent EV pathway in SOCS3 release.

3) The authors claim that their assay using crosslinkable CFSE provides a specific means for measuring packaging of cytosolic cargoes into EVs. However, I would say that once CFSE gets in the cell and is processed (by intracellular esterases), it should also label transmembrane proteins found on the many membrane compartments of the cell (ER, Golgi, endosomes, mitochondria etc.), even at the plasma membrane (cytosolic tails/domains of membrane receptors and channels). Thus, since many of these membrane proteins are likely to be recruited to the many classes of EVs released by the cell, the assay here described does not exclusively/specifically monitor release of cytoplasmic cargoes. Please address accordingly.

4) Treatment with CSE, H2O2, MG132, Bortezomib increased extracellular levels of SOCS3. Is this specific for SOCS3? Can the authors provide a positive control cytoplasmic cargo (perhaps TSG101, which has been described to regulate microvesicle formation) that also gets exported under these conditions? And similarly, Can the authors provide a negative control cytoplasmic cargo that does not get exported under these conditions? For example, if AMs are transfected with GFP, is cytosolic GFP going to be more packaged/exported upon these treatments? This is conceptually important, since it would provide evidence for distinguishing a passive (apoptotic bodies) from an active (EV biogenesis and release) mechanism of export. Thus, altogether in Fig 3D and Fig6D I would rather like to see additional EV markers and the above-mentioned control cargoes.

With regards to these treatments: Intensity of lower molecular-weight species of SOCS3 appears to increase as the MG132, Bortezomib or H2O2 doses increase. I wonder if these are cleaved forms of SOCS3 protein in response to caspase-mediated cleavage/Apoptosis?

5) The authors speculate that SOCS3 is packaged in plasma membrane-derived EVs, in a similar way as the previously described arrestin domain-containing protein 1 (ARRDC1)-mediated microvesicles (ARMMs). Yet, they do not provide any evidence that SOCS3 is exported by these types of vesicles. Moreover, I do not understand the relationship between proteasomal activity and recruitment of cargoes to microvesicles. Members of the ESCRT machinery (including TSG101) recruit ubiquitinated cargoes to both endosomal intraluminal vesicles but also to plasma membrane-derived EVs. In this scenario, ubiquitination of cargoes has nothing to do with their targeting to proteasome-dependent degradation.

Thus, to me it is not clear in the manuscript (lines 325-330, also in Abstract) the link between proteasomal degradation and cargo recruitment to a specific type of EV. Unless, what the authors mean is that SOCS3 is normally degraded in the proteasome, and inhibition of the proteasome increases the levels of SOCS3 and its release? In Fig 6A authors claim that the levels of SOCS3 do not change upon proteasomal function inhibition. I do see an increase in the lower molecular weight species of SOCS3. However, in order to demonstrate if proteasome inhibition does not interfere with the half life of intracellular SOCS3, the authors would have to show, in a time course experiment (with cycloheximide), that the levels of SOCS3 do not change.

Minor points:

1) In lane 293 and 295, Large EVs (lEVs) should be indicated not only as an abbreviation. Also small EVs (sEV).

2) The authors should briefly mention in the manuscript what is the DCFDA assay (used to measured intracellular ROS levels) based on (this would help the non-expert reader).

3) The authors should not exclude the possibility that altering the levels of ROS in the cell also impacts autophagy, which may also represent an unconventional mechanism of export for cytoplasmic proteins.

4) Please cite this paper (PMID: 26858453) accordingly in the manuscript.

Round 2

Reviewer 3 Report

I do understand about the current situation (COVID-19 pandemic) in getting in the Lab/do experiments, and I appreciate that the authors discuss (in the manuscript) the limitations of not having shown EV markers in none of their experiments.

Regarding Authors’ reply to Major point #2:

An alternative to using siRNA could be Lentiviral infections of shRNA-based vectors.

Also, the Neutral sphingomyelinase inhibitor (GW4869) has been widely used in the EV field, and while it is true that it may also have broad effects, it has been shown to clearly shut off EV release in multiple contexts and cell lines.

Nevertheless, the authors now report presence of Vps4 in their EV fractions. Given that Vps4 is a ESCRT component, the authors should mention instead “ESCRT-dependent EV biogenesis” (line 266 of the current manuscript version).

Regarding Authors’ reply to Major point #4:

Due to the lack of a negative control protein (i.e. GFP or other cytosolic protein not selectively targeted in the SOCS3-EV population) the authors need to emphasize more in the manuscript that they cannot discard that SOCS3 is passively and not actively targeted into these EVs. They show a similar trend of recruitment for Vps4, but without a cytosolic protein protein that shows the opposite or no increased enrichment (upon treatment with oxidative agents etc..) they cannot fully conclude that SOCS3 recruitment is selective. In other words, I would be very concerned if all cytosolic proteins are equally recruited to these EVs as SOCS3.

Regarding Authors’ reply to Major point #5:

First of all, ARRDC1 is NOT a chaperone protein, but more precisely a substrate adaptor of Nedd4-ubiquitin Ligases, which means that ARDDS1 normally binds substrates targeted for ubiquitination mediated by Nedd4-ubiquitin ligases. Whether an ARRDC1-Nedd4-HECT ligase complex ubiquitinates and recruits substrates/cargoes to EVs remains unknown. Yet, it has nevertheless been shown that Ndfip1 (another Nedd4 E3 ligase adaptor) binds and mediates PTEN poly-ubiquitination, ultimately promoting PTEN loading into EVs (PMID: 23012657).

Also, ARRD1C does NOT require ubiquitination for its targeting to the PM, but rather its arrestin-like domain. The authors should carefully read Nabhan et al.(2012) PNAS119:4146. In this paper, ARRD1C was found to be poly-ubiquitinated by the HECT E3 Ligase WW2, and it was proposed that this modification increases ARRMs release. Moreover, this modification was not shown to be required for interaction with the ESCRT member TSG101, nor it was shown to be required for recruiting any cargoes (including NOTCH2 via ITCH > PMID: 28955033). To my knowledge, ARRD1C poly-ubiquitination seemed to increase ARMMs release, but nevertheless, it is still unclear what is the role of such modification in the context of EV cargo recruitment.

On the other hand, not all poly-ubiquitin chains signal for proteasomal degradation. Usually K48 or K11-linked Ubi chains target substrates for proteasomal destruction, but other chains (K63 or K33) serve as regulatory signals in a broad range of biological processes including regulation of membrane trafficking or innate immunity. Thus, disrupting the proteasomal activity will affect substrates targeted for proteasomal degradation (bearing K48-linked Ubi chains) but it will NOT affect the dynamics of many other proteins that are also poly-ubiquitinated with other types of Ubi-chains (other than K48-chains, for example).

Finally, long-term inhibition of the proteasome (using MG132 or Bortezomib) can lead to exhaustion of “free” ubiquitin monomers and therefore inhibit ubiquitination due to a “lack” of ubiquitin-chain building blocks. This is because the proteasome, apart from degrading substrates, they also recycle Ubiquitin for other rounds of ubiquitination.

Altogether, according to what I mentioned above the authors should make an effort to better clarify the connection between proteasomal inhibition and SOCS3 release (please improve explanation in paragraph between lines 337 and 342 in the manuscript). In my opinion, the authors should more precisely explain what these findings mean at the biological level.
